# Study of Joint Symmetry in Gait Evolution for Quadrupedal Robots Using a Neural Network

Zainullah Khan [1], Farhat Naseer [1], Yousuf Khan [1,*], Muhammad Bilal [2] and Muhammad A. Butt [3,4]

1. Embedded Systems Research Group, Department of Electronic Engineering, Balochistan University of Information Technology, Engineering and Management Sciences, Quetta 87300, Pakistan; zain.9496@gmail.com (Z.K.); farhaty16@gmail.com (F.N.)
2. Department of Telecommunication Engineering, Balochistan University of IT, Engineering and Management Sciences, Quetta 87300, Pakistan; muhammad.bilal4@buitms.edu.pk
3. Institute of Microelectronics and Optoelectronics, Warsaw University of Technology, Koszykowa 75, 00-662 Warszawa, Poland; ali.butt@pw.edu.pl
4. Samara National Research University, 443086 Samara, Russia
* Correspondence: yousuf.khan@buitms.edu.pk

**Abstract:** Bio-inspired legged robots have the potential to traverse uneven terrains in a very efficient way. The effectiveness of the robot gait depends on the joint symmetry of the robot; variations in joint symmetries can result in different types of gaits suitable for different scenarios. In the literature, symmetric and asymmetric gaits have been synthesized for legged robots; however, no relation between the gait effectiveness and joint symmetry has been studied. In this research work, the effect of joint symmetry on the robot gait is studied. To test the suggested algorithm, spider-like robot morphology was created in a simulator. The simulation environment was set to a flat surface where the robots could be tested. The simulations were performed on the PyroSim software platform, a physics engine built on top of the Open Dynamics Engine. The quadrupedal robot was created with eight joints, and it is controlled using an artificial neural network. The artificial neural network was optimized using a genetic algorithm. Different robot symmetries were tested, i.e., diagonal joint symmetry, diagonal joint reverse symmetry, adjacent joint symmetry, adjacent joint reverse symmetry and random joint symmetry or joint asymmetry. The robot controllers for each joint symmetry were evolved for a set number of generations and the robot controllers were evaluated using a fitness function that we designed. Our results showed that symmetry in joint movement could help in generating optimal gaits for our test terrain, and joint symmetry produced gaits that were already present in nature. Moreover, our results also showed that certain joint symmetries tended to perform better than others in terms of stability, speed, and distance traveled.

**Keywords:** quadrupedal robot; genetic algorithm; gait evolution; neural networks; robot morphology; robot generations



## 1. Introduction

Legged robots (that have been inspired by the morphology of real animals) can traverse uneven terrains in a more efficient way than their wheeled counterparts [1,2]. Among legged robots, quadrupedal robots are the most agile and the most stable [3–5].This motivates to mimic nature and thereby adapting robots and vehicles to suit uneven terrains rather than adapting the environment to cater to the needs of wheeled vehicle. However, such legged robot morphologies can prove to be quite a challenge to control manually [6]; therefore, it is very common to automate the process of gait generation for such robots. A robot gait is the movement of actuators that allows the robot to traverse an area [7]. The gaits for quadrupedal robots are categorized into two types according to [8,9]: the first type has joint symmetry and the second type has joint asymmetry. Joint symmetry occurs when the movement of one joint is replicated by another joint in the robot, and joint asymmetry

occurs when the movement of two is completely different from one another, i.e., there is no correlation between the movement of one joint with the movement of another joint.

Quadrupedal animals in nature possess both aforementioned gait varieties. The three most common quadrupedal robot gaits are walking, trotting and galloping [10]. Out of these, the walking gait is asymmetric, and the movement of all the legs is different from one another [11]. On the other hand, the trot and gallop gaits are symmetric. In the trot gait, the diagonal legs of a quadruped are in symmetry [12], while in the gallop gait, the front and rear leg pairs are in symmetry [13]. It should also be noted that the trot and gallop gaits are used for running and, hence, among these mentioned gaits, running gaits are symmetric while walking gaits are asymmetric [14].

Inspired by nature, multiple approaches have been taken to synthesize gaits for quadrupedal robots. In [15,16], a static gait for a quadrupedal robot is developed that allows it to traverse uneven terrains; the gait is asymmetric and the robot is controlled using a path planning algorithm. The robot can successfully traverse a rough terrain. Contrary to this, refs. [17–19] develop gaits for quadrupedal robots with joint symmetry. Trotting gaits are developed in simulation, and then later the same gaits are tested on a physical robot. The results show that the robot can traverse plain areas. In [20–22], running and turning gaits are synthesized for an under-actuated quadrupedal robot; the gallop gait is chosen for running, making the running gait symmetric, and the turning gait is also symmetric.

All the methods stated above allow the robots to successfully traverse different terrains; however, the gaits are all fixed and cannot change if the terrain is slightly altered. The robots have no means of adjusting their gaits to match the environment that they are in. Therefore, to adapt well to the environment, the gaits must be synthesized using an optimization process. Some of the most common gait optimization methods used in the literature are the genetic algorithm (GA) [23] and artificial neural networks (ANNs) [24].

As stated above, ANNs and the GA are often used to control robot gaits. The works in [25–28] utilize the GA to evolve gaits for quadrupedal robots that are controlled by ANNs. In [25,28] a comparative study is performed between different types of ANNs to determine which one will produce a gait that is better than the rest of the ANNs. It was evident from their result that the gaits that resulted in symmetric movement often performed the best when their utility function favored speed; however, when faced with uneven terrains, the symmetric gaits performed badly and the asymmetric gaits performed well. Similarly, in [26], gaits were evolved for a quadrupedal robot, and the entire process was performed on a hardware setup. The robot was configured in such a way that its joints were symmetric about the center of the robot. The evolved gait performed very well and the robot successfully traversed even surfaces. However, evolving gaits on a physical robot is a slow process and it may take hundreds of hours to reach an optimal gait. The work in [27] is an extension of [26], and it still employs the same joint symmetry; however, the first few generations of the robot are simulated and, when the robot learns to walk properly, the controllers are transferred to a hardware robot. This saves the time it takes to evolve gaits for a complete hardware robot setup, and the loss of electronic components is also avoided.

The studies mentioned above demonstrate how joint symmetry and asymmetry aid in robot locomotion in different environments and on different terrains; however, no critical analysis has been carried out that relates the effects of joint symmetry to a robot's ability to traverse its environment. Therefore, in this research, we contribute to the field by experimenting with different joint symmetries and studying their effect on the ability of a robot to traverse a flat terrain. We will be using the GA to optimize an ANN controller for our robot. We have also designed a custom fitness function which allows us to evaluate the robot controllers. The results of this study will help in determining whether joint symmetry aids in robot movement or not; moreover, this study also aims to create a benchmark that can help future researchers in deciding which joint symmetries to choose for their robots.

The rest of the paper is arranged in the following order: Section 2 will discuss our methodology; our simulation will be detailed in Section 2.1; our controller will be discussed in Section 2.2; our optimization algorithm will be discussed in Section 2.3; and in Section 2.4 our selection criteria will be overviewed. Finally, the results will be discussed in Section 3.

## 2. Methodology

The simulations were performed on the PyroSim [29] software platform, a physics engine built on top of an Open Dynamics Engine [30]. To test the suggested algorithm, robot morphology was created in the simulator. Ten individual robots were created in the simulator: the so-called test-population. Every individual in the population was controlled by a fully connected neural network. The designed neural network-based controller could evolve itself using the GA. The robot morphology remained the same for each joint symmetry that was experimented on; however, the controller (ANN) was changed slightly each time to form different joint symmetries. A detailed explanation for all the steps involved in evolving the controllers can be found in Figure 1.

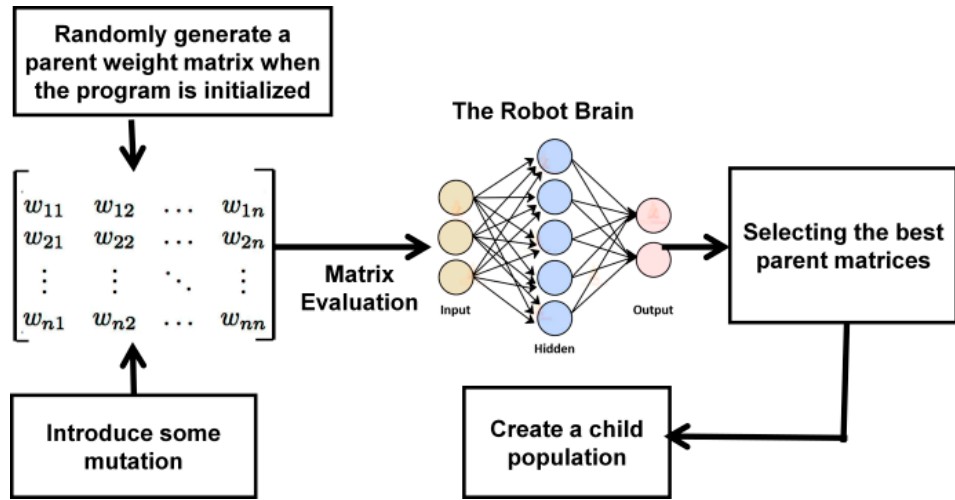

**Figure 1.** The block diagram of the controller evolution.

### 2.1. Simulation

As mentioned above, the simulations were performed in PyroSim. The robot body was created using geometric shapes, and a cuboid was used to make the main body while the limbs were made from cylindrical shapes. The robot body was completed by joining all the shapes through hinge joints, hence making a degree of freedom. Each robot leg had two joints, one for the hip and the second for the knee. A total of 4 limbs and 8 joints were created, as shown in Figure 2. The simulation environment was set specifically to suit the environment where the physical robot would be used, and the parameters are given in Table 1.

**Table 1.** Simulation parameters used in the simulator.

| Parameter | Symbol | Value |
|---|---|---|
| Robot body length | $l$ | 0.2 m |
| Robot body width | $w$ | 0.2 m |
| Robot body height | $h$ | 0.05 m |
| Length of cylinder | $cL$ | 0.2 m |
| Radius of cylinder | $cR$ | 0.02 m |
| Mass of each robot body part in the simulation | $m$ | 1 kg |

**Table 1.** *Cont.*

| Parameter | Symbol | Value |
|---|---|---|
| Gravity | $g$ | $-9.8 \text{ ms}^{-2}$ |
| Number of joints | $J$ | 8 |
| Number of motors | $M$ | 8 |
| Motor impulse | $\tau$ | 0.15 |
| Simulation world step time | $dt$ | 0.05 |
| Total number of timesteps for the simulation | $T$ | 1000 |
| ANN recall interval timesteps | $Rc$ | 60 |
| ANN inputs | $I$ | 9 |
| ANN outputs | $O$ | 8 |
| Number of individuals in the population | $P$ | 10 |
| Number of generations | $G$ | 200 |

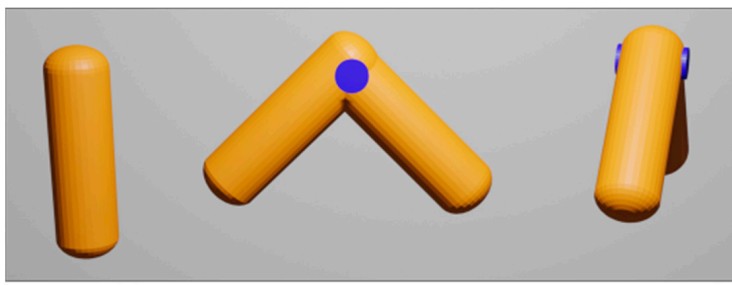

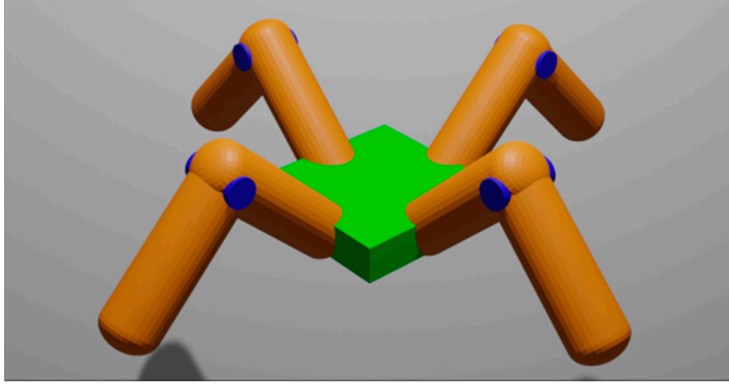

**Figure 2.** The figure shows the single components that form the robot. The single segment is represented by a capsule shape and the joining two of these capsules forms a joint. The figure at the bottom shows these jointed segments further joining to a cube-shaped body.

*2.2. Controller*

As stated earlier, the controller used in this study was an ANN. ANNs are mathematical models of the biological brain, and they are used to imitate the nervous system of biological beings. The ANN controls the robot using a feedforward neural network with a hyperbolic tangent activation function. The neural network has 9 inputs; the number of outputs is determined by the joint symmetry used and will be further discussed in this section. The first eight inputs of the neural network are the proprioceptive sensor values that return the current joint positions of the robot, and the ninth input is a bias neuron [31]. The data are passed through the neural network, the values of the output neurons are generated in the form of new joint positions, and the robot joints are actuated according to the joint positions. We did not include any hidden layers to keep the solution space small. The neural network was called once every 60 timesteps to avoid any jerky moment.

The equation below gives the output of the neural network:

$$O_j = \tanh \sum_{i=0}^{l} w_{ij} J_j \tag{1}$$

where $O_j$ is the output value for joint $j$, $w_{ij}$ represents the weight that connects input neuron $i$ with output neuron $j$, and $J_j$ is the current joint position of joint $j$.

To create different joint symmetries, the robot morphology is not altered; however, the controller was altered slightly to form the following symmetries.

### 2.2.1. Diagonal Joint Symmetry

The diagonal robot legs shared the same joint angles in this symmetry. The ANN controller, in this case, had 4 outputs, relating to 4 joint values. The ANN output angles were assigned to the two front legs of the robot and the same angle values were used for the legs that were diagonal to the front legs, i.e., the joint values of the front left leg were duplicated to the joint values of the hind right legs, as seen in Figure 3b.

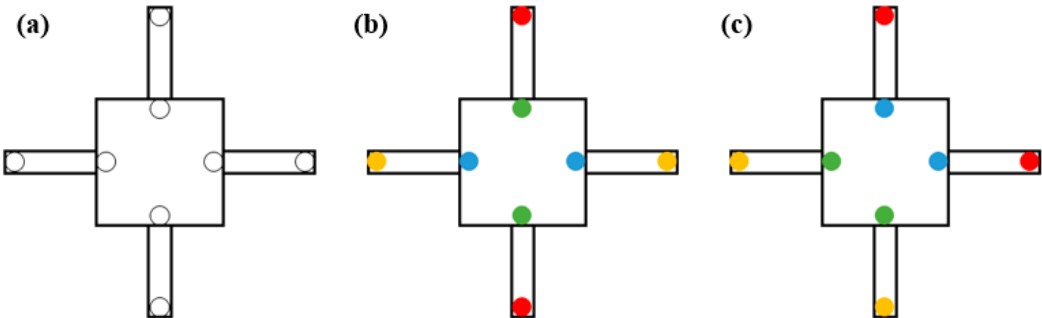

**Figure 3.** Top view of the quadrupedal robot. (**a**) Joint asymmetry in the quadrupedal robot. (**b**) Diagonal joint symmetry is shown in the quadrupedal robot. (**c**) Adjacent joint symmetry is shown in the robot.

### 2.2.2. Adjacent Joint Symmetry

For this joint symmetry, the ANN architecture still had 4 outputs; however, the leg symmetry was changed. The ANN output joint values were applied to the left front and left hind leg and the symmetry was formed between the front two legs and the hind two legs. Therefore, the joint values of the front left leg were replicated to the front right leg and the joint values of the hind left leg were replicated to the hind right leg, as seen in Figure 3c.

### 2.2.3. Diagonal Joint Reverse Symmetry

This joint symmetry was similar to the diagonal joint symmetry; the controller architecture remained the same (4 outputs), the only difference, in this case, was that when the joint values were replicated they were multiplied by −1, which made the diagonal legs move 180° out of phase.

### 2.2.4. Adjacent Joint Reverse Symmetry

The symmetric joints and the ANN controller remained the same as in the adjacent joint symmetry; however, in this case, the joint values were multiplied by −1 when they were replicated, shifting the phase of the symmetric legs by 180°.

### 2.2.5. Joint Asymmetry or Random Joint Movement

The joints did not form any symmetry: every joint moved in an arbitrary direction. This is shown in Figure 3a.

### 2.3. Algorithm

As mentioned before, we used the GA to optimize the neural network. The main steps of our proposed algorithm are shown in Figure 4.

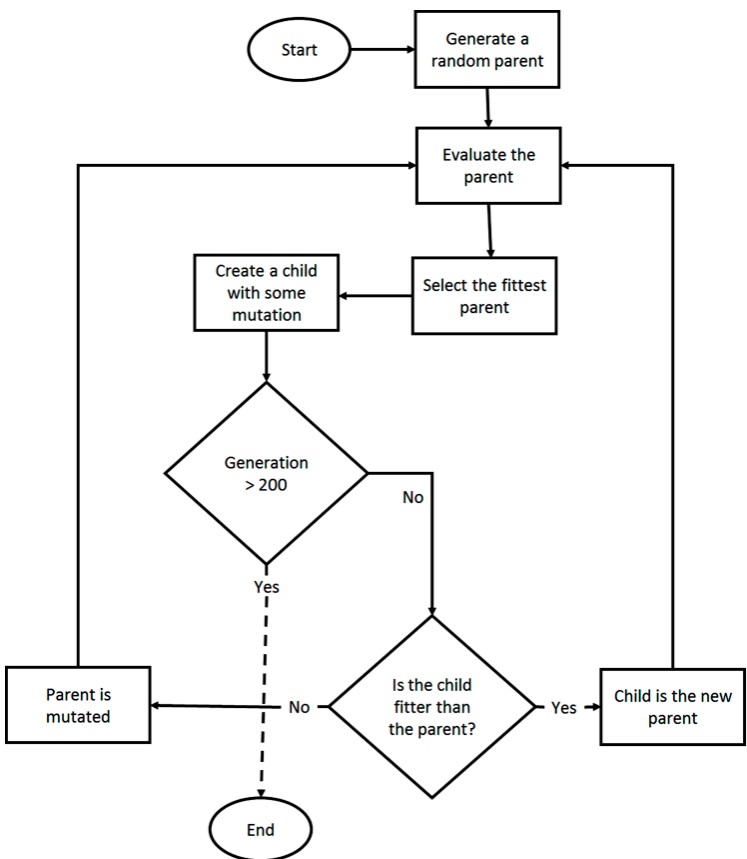

**Figure 4.** Flowchart of the algorithm used in this paper.

The simulator and the GA were coded in python. The simulation was run for 10 individual robots at a time where each robot had a random weight matrix that allowed the neural network to control the robot, and each simulation lasted for 1000 timesteps. When the simulation ended, the GA selected the best-performing individual based on the selection criteria, as mentioned in the next section. This individual was allowed to make copies of its weight matrix and the weight matrix of each copy was slightly changed to introduce some variation that would allow the child to perform slightly different from the parent. The different behavior could result in a child performing better than the parent; in such a case, the child becomes the new parent and the process continues until a set number of generations is completed, which in this case was 200.

*2.4. Selection Criteria*

After one simulation cycle was terminated, the robots from all the simulations were evaluated and the best performing controllers were selected using a mathematical function that was user-defined. This function was called the fitness function and is provided below:

$$F = (\sqrt{x^2 + y^2})(\sum_{t=1}^{T} \sum_{j=0}^{J} J_{tj} - J_{(t-1)j})(1 - TS) \qquad (2)$$

where $x$ and $y$ are the distances traveled by the robot in the $x$ and $y$ directions, respectively. $J_{tj}$ and $J_{(t-1)}$ represent the positions of joint $j$ at time $t$ and $t - 1$, respectively. $TS$ is the value of a touch sensor attached to the back of the robot's body, such that $TS$ remains one '1' when the robot flips itself over.

## 3. Results and Discussion

Once the simulations ended, the results from all the joint symmetries were analyzed. This section will deal with displaying and discussing the results obtained.

### 3.1. Adjacent Joint Symmetry and Reverse Symmetry

The result of the adjacent joint symmetry is shown in Figure 5. For two trials, the robot did not score a high fitness value; however, for the rest of the trials the robot scored a high value, showing that the robot could travel a notable amount of distance. This symmetry made the robot very stable, and the robot was able to gallop. This can be used to develop galloping gaits for robots.

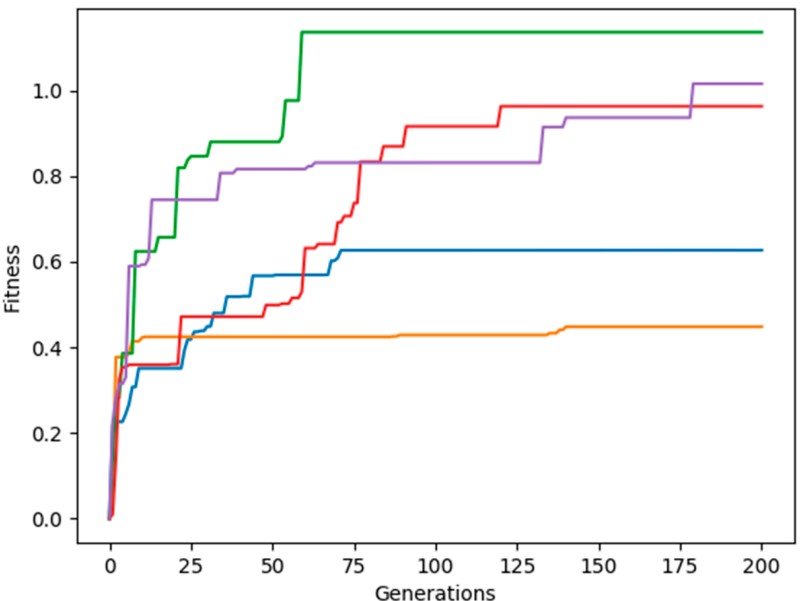

**Figure 5.** Adjacent joint symmetry fitness for five simulation runs, showing that the majority of the robots reached a high fitness value.

The fitness values for five evolution trials of adjacent joint reverse symmetry are shown in Figure 6. Note that most of the robots achieved high fitness values. This gait exhibited a jerky robot movement and sent the robot in arbitrary directions occasionally. However, this joint symmetry achieved the highest fitness values on average.

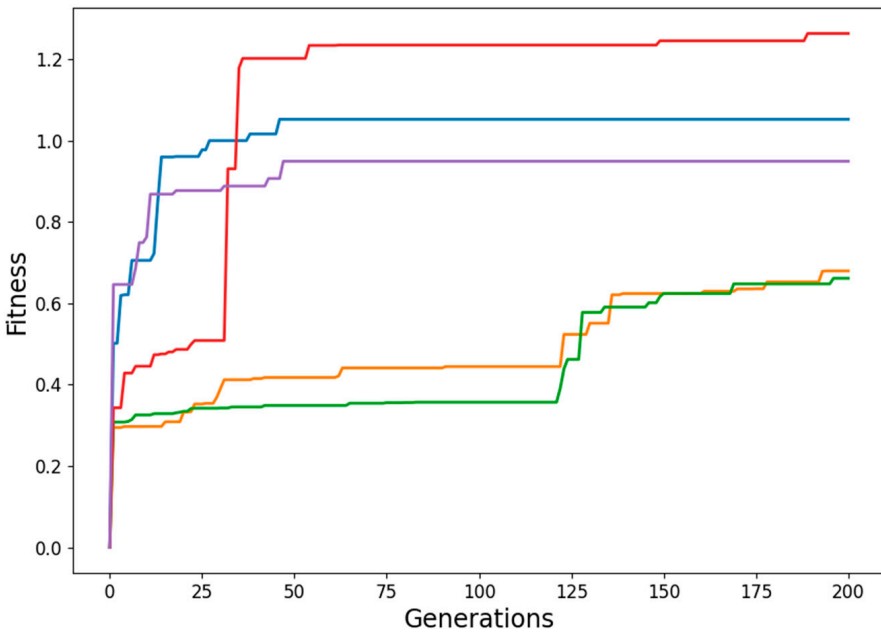

**Figure 6.** Adjacent joint reverse symmetry fitness for 5 simulation runs, showing that the majority of the robots reached a high fitness value.

### 3.2. Diagonal Joint Symmetry and Reverse Symmetry

The result for diagonal joint symmetry is shown in Figure 7. This joint symmetry had the least fitness score per evolution run, as well as on average. This is because when the diagonal joints were symmetrical, the forces applied by the robot's leg had the same magnitude but the direction was opposite, causing the forces to cancel each other out. Consequently, the robot stayed in the same spot.

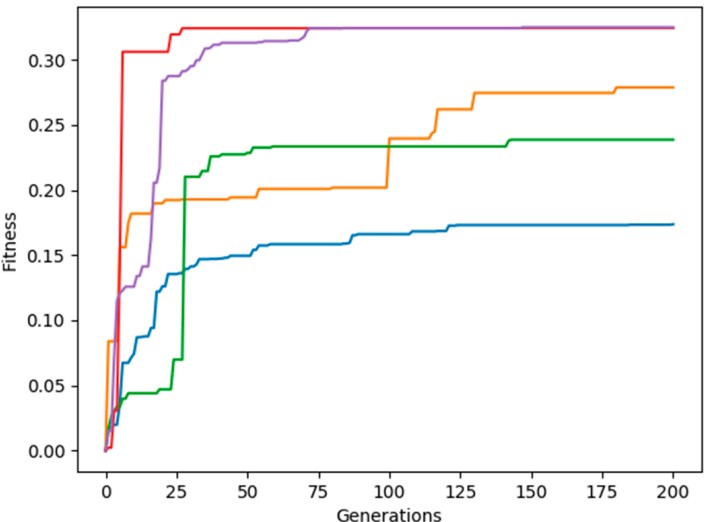

**Figure 7.** Diagonal joint symmetry fitness for 5 simulation runs.

The fitness graph for diagonal reverse symmetry is shown in Figure 8. The maximum fitness score and the average fitness score of this joint symmetry was the second highest. This high fitness value is attributed to the robot's joint movements that allowed it to walk stably by taking turns in using diagonal leg pairs to locomote. The gaits produced with this method resembled the creep gait, which is very common in nature.

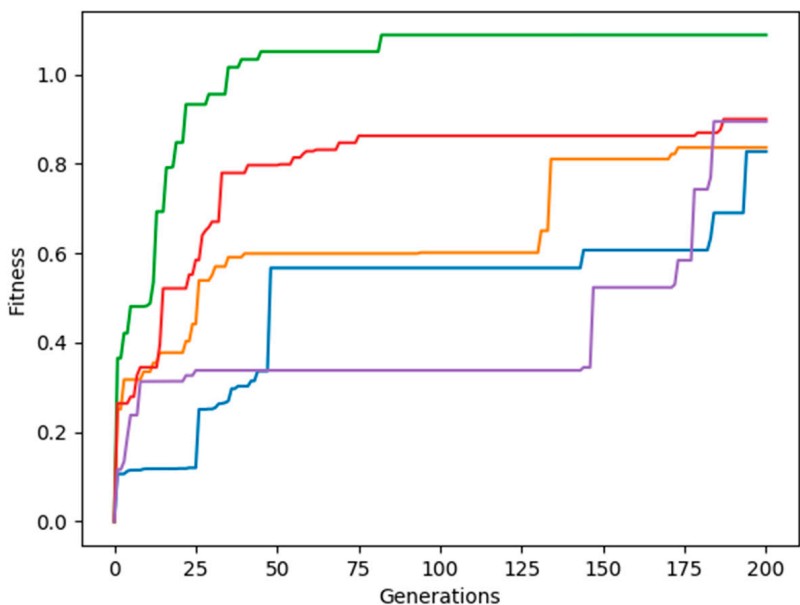

**Figure 8.** Diagonal joint reverse symmetry fitness for 5 simulation runs.

### 3.3. Random Joint Movement

The fitness for random joint movement is shown in Figure 9. The fitness values achieved by the robot were all caused by jerky movements that threw the robot around,

causing the robot to gain a high fitness score. This robot gait was the least stable and produced gaits that were not optimal for a flat terrain. However, this random joint movement could be helpful over uneven terrains.

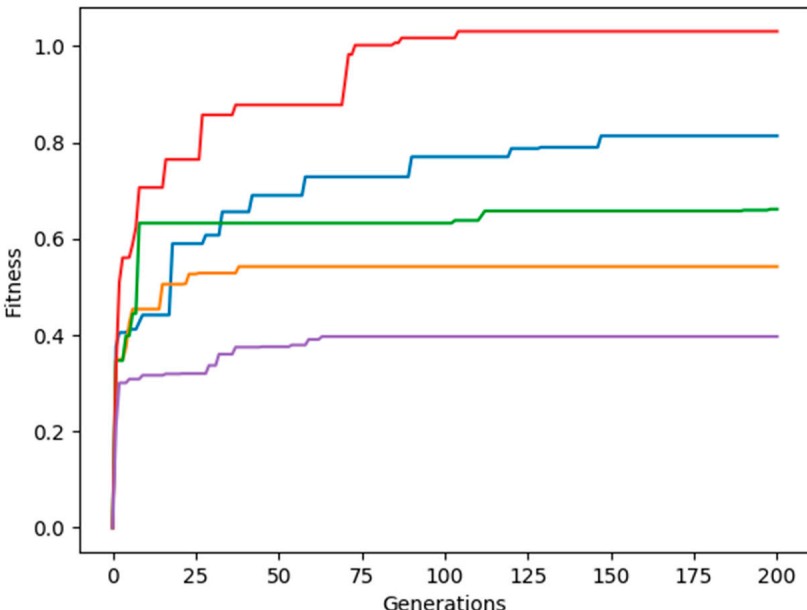

**Figure 9.** Random joint movement or joint asymmetry for 5 different simulation runs.

Figure 10 shows the average fitness scores of all the joint symmetries. As discussed above, the joints with adjacent reverse symmetry and diagonal reverse symmetry achieved the highest fitness scores, on average. The adjacent joint symmetry produced stable gaits that resembled a galloping motion, which could be useful in developing high-speed gaits. The gaits produced with adjacent joint reverse symmetry were able to generate the highest fitness score; however, these gaits were unstable and they would send the robot in arbitrary directions, which led to a high fitness score and no usable gaits. The diagonal joint symmetry produced the lowest fitness scores because the robot would stay at the origin, and the gaits produced were not useful at all in propelling the robot. The diagonal joint reverse symmetry produced useful and stable movements that resembled the creep gait. The robot was able to traverse the flat terrain with ease with this joint symmetry. Moreover, the gait speed was slow, which further made the robot more stable. The robot with no joint symmetry was able to achieve high fitness values and the robot mostly produced a jerky movement that would move the robot in arbitrary directions. Although the gait was useless for traversing a flat terrain, it could prove useful in moving over uneven terrain or an environment with obstacles.

Figure 11 shows a sequence of frames that show the robot movement in the simulator for all the joint symmetries used. Figure 11a shows the adjacent reverse symmetry; the robot travelled far, however, the robot body was very jerky and threw itself around. Figure 11b shows that the gaits produced by adjacent symmetry could jump high and were very stable. Figure 11c shows the diagonal joint reverse symmetry. The gaits produced mimicked nature, and they were the most stable gaits produced. Figure 11d shows the diagonal joint symmetry; in this case, the robot stayed near the origin. Finally, Figure 11e shows the robot with no joint symmetry. The robot was thrown around to an arbitrary position.

The importance of leg symmetry in gait generation has been highlighted in our research and our results agree with the literature that we reviewed. The gaits produced by [17–19] were symmetric and produced the best results for the authors. The gait that they generated was the trotting gait. Similarly, the authors in [20–22,32] produced galloping gaits that were also symmetric, and our experiments proved that the galloping gaits could be very fast and stable when we developed these gaits with adjacent joint symmetry. Finally, our

results showed that gaits with no symmetry are not optimal for flat terrains due to the high variability in the joint movements; therefore, we also conclude that random joint movements are more suitable for uneven terrains, which agrees with [15,16,33].

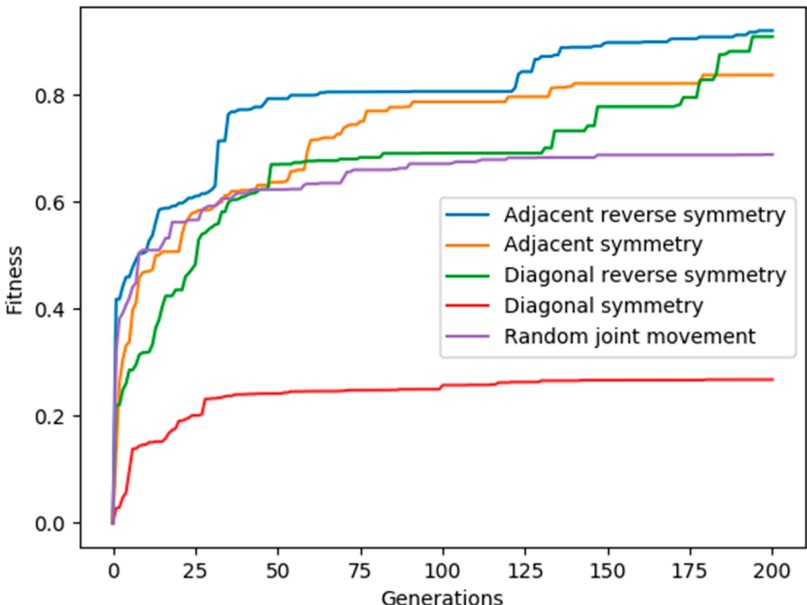

**Figure 10.** Average fitness values of all the joint symmetries and asymmetries used.

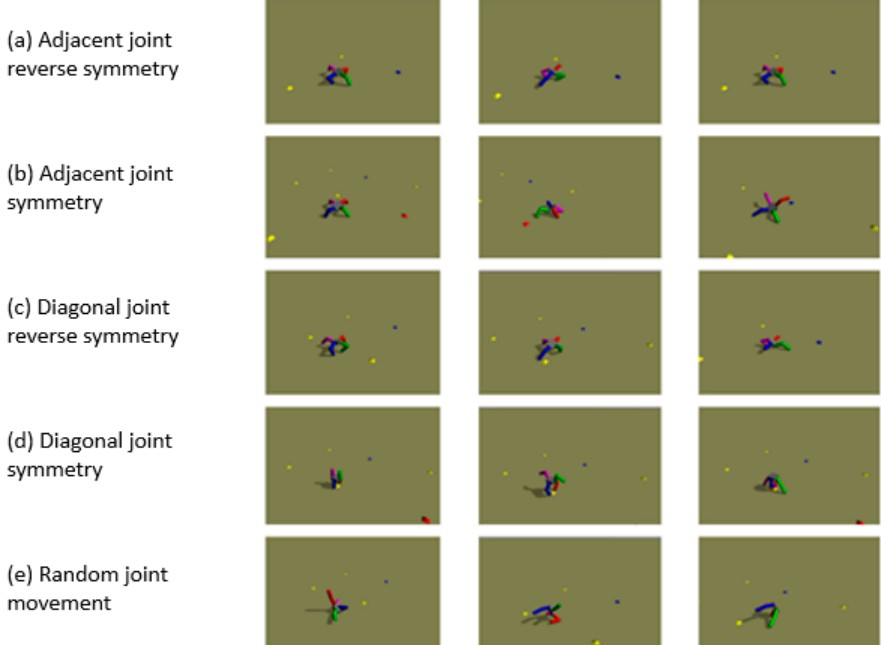

**Figure 11.** Robot movements in the simulator are based on the symmetry used. (**a**) shows the adjacent reverse symmetry. (**b**) shows the adjacent symmetry. (**c**) shows the diagonal reverse symmetry. (**d**) shows diagonal symmetry, and (**e**) shows random joint movement.

Our results give an idea about the joint symmetries that can produce the best results for a quadrupedal robot. The results showed that the wrong joint symmetry can result in unstable gaits that can damage a robot in the real world; moreover, the wrong joint symmetry can produce gaits that are suboptimal for a given terrain. In contrast to this, the right joint symmetry can produce agile and stable gaits that are optimal for a given terrain.

## 4. Conclusions

This paper performed a novel study to determine whether joint symmetry in quadrupedal robots could improve their locomotion gaits. The results showed that joint symmetry can help robots to evolve stable gaits. Our results showed that adjacent joint symmetry produces the fastest gaits, while the diagonal joint reverse symmetry produced gaits that were the slowest; however, both gaits exhibited qualities of gaits found in nature and were stable. The rest of the joint symmetries produced gaits that were not optimal and unstable. Moreover, symmetric gaits can travel greater distances. Therefore, choosing the right type of joint symmetry is important when generating robot gaits. This study can be further expanded to examine the effect of joint symmetry on a physical robot. It will also be interesting to experiment with evolving symmetric and asymmetric gaits for other robot morphologies.

**Author Contributions:** Conceptualization, Z.K., F.N. and Y.K.; methodology, Y.K., M.B. and M.A.B.; software, Z.K. and F.N.; validation, Y.K., M.B. and M.A.B.; formal analysis, Y.K. and M.B.; investigation, Z.K., Y.K., F.N. and M.B.; resources, Y.K., M.B. and M.A.B.; data curation, Y.K. and M.A.B.; writing—original draft preparation, Z.K., Y.K., F.N. and M.B.; writing—review and editing, Y.K., M.B. and M.A.B.; visualization, Y.K. and M.A.B.; supervision, Y.K, M.B. and M.A.B.; project administration, Y.K. and M.A.B.; funding acquisition, Y.K. and M.A.B. All authors have read and agreed to the published version of the manuscript.

**Funding:** The authors would like to thank the Higher Education Commission of Pakistan for research grant No. 21-2129 under Startup Research Grant Program (SRGP) which made this work possible. Moreover, the authors are also grateful to Ignite National Grassroots ICT Research Initiative (NGIRI) for granting final year project funding for this project.

**Institutional Review Board Statement:** Not appliable.

**Informed Consent Statement:** Not appliable.

**Data Availability Statement:** Not appliable.

**Acknowledgments:** The authors are thankful to Embedded Systems research group at BUITEMS Quetta for their valuable support to the project.

**Conflicts of Interest:** The authors declare no conflict of interests.

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
