# Peer review of "Study of Joint Symmetry in Gait Evolution for Quadrupedal Robots Using a Neural Network"

_technologies, doi:10.3390/technologies10030064_

Round 1

Reviewer 1 Report

The proposed subject in the paper is quite timily and the topic is familiar to the reviewer. Therefore globally the ideas proposed by the authors are innovative in some sense.

The paper is sutable for the journal. The theory performed by the authors for theisr problem, is in agreement with a general one regarding the imperfection theory. Therefore It is suggested to remark this concept and to cite the following key point paper.    

AccessOpen AccessVolume 9, Pages 29573 - 295832021 Article number 9351920 Document type Article• Gold Open Access Source type Journal ISSN 21693536 DOI 10.1109/ACCESS.2021.3058506 View more   

Imperfections in Integrated Devices Allow the Emergence of Unexpected Strange Attractors in Electronic Circuits

Moreover due to the further comments required it is interesting to havve a view as the control proposed in a single robot could be viewed in the context of collective behaviour. I therefore suggest to take as reference the following contribution that must be included as futher references.

Dynamical network interactions in distributed control of robots

Author Response

Thank you for the suggestions and positive criticism. We have revised the manuscript as per the reviewer's suggestions. Please see the attached file. 

Reviewer 2 Report

The article deals with the study of the symmetry effects of the locomotion system of biologically inspired mobile robots. The topic of the work is current and has the potential for novelty. I recommend that you modify the abstract of the article to better capture the essence of the article. I recommend the authors of the article to describe the specific benefits of their research. In the end, the result and potential risks are not clearly presented, so I recommend adding a discussion chapter where these questions will be answered. Figure 11 has a low informative value.

Author Response

(The authors gave the same response as above.)

Reviewer 3 Report

This paper presents the effect of joint symmetry on the robot’s gait. The paper is interesting, and clearly presents the methodology and the results. However, there are a lot of linguistic errors, and the results are not discussed with the literature. As far I know, there are more recent similar studies that are not presented in this paper. The major lack of this paper is the inexistence of the comparison of this study with the literature. It must be corrected before acceptance.

Author Response

(The authors gave the same response as above.)

Round 2

Reviewer 3 Report

The manuscript is improved, and it can be accepted.